# Cyclic GMP in Liver Cirrhosis—Role in Pathophysiology of Portal Hypertension and Therapeutic Implications

**DOI:** 10.3390/ijms221910372

**Published:** 2021-09-26

**Authors:** Wolfgang Kreisel, Adhara Lazaro, Jonel Trebicka, Markus Grosse Perdekamp, Annette Schmitt-Graeff, Peter Deibert

**Affiliations:** 1Department of Medicine II, Gastroenterology, Hepatology, Endocrinology, and Infectious Diseases, Faculty of Medicine, Medical Center, University of Freiburg, 79106 Freiburg, Germany; 2Institute for Exercise and Occupational Medicine, Faculty of Medicine, Medical Center, University of Freiburg, 79106 Freiburg, Germany; adhara.lazaro@alumni.uni-heidelberg.de (A.L.); peter.deibert@uniklinik-freiburg.de (P.D.); 3Translational Hepatology, Department of Internal Medicine I, Goethe University Clinic Frankfurt, 60590 Frankfurt, Germany; jonel.trebicka@kgu.de; 4Institute of Forensic Medicine, Faculty of Medicine, Medical Center, University of Freiburg, 79106 Freiburg, Germany; markus.perdekamp@uniklinik-freiburg.de; 5Faculty of Medicine, Medical Center, University of Freiburg, 79106 Freiburg, Germany; annette.schmitt-graeff@uniklinik-freiburg.de

**Keywords:** liver cirrhosis, liver fibrosis, portal hypertension, NO-cGMP pathway, sGC, PDE-5, cGMP, nitric oxide, sGC modulators, PDE-5 inhibitors, hepatic encephalopathy, plasma markers

## Abstract

The NO-cGMP signal transduction pathway plays a crucial role in tone regulation in hepatic sinusoids and peripheral blood vessels. In a cirrhotic liver, the key enzymes endothelial NO synthase (eNOS), soluble guanylate cyclase (sGC), and phosphodiesterase-5 (PDE-5) are overexpressed, leading to decreased cyclic guanosine-monophosphate (cGMP). This results in constriction of hepatic sinusoids, contributing about 30% of portal pressure. In contrast, in peripheral arteries, dilation prevails with excess cGMP due to low PDE-5. Both effects eventually lead to circulatory dysfunction in progressed liver cirrhosis. The conventional view of portal hypertension (PH) pathophysiology has been described using the “NO-paradox”, referring to reduced NO availability inside the liver and elevated NO production in the peripheral systemic circulation. However, recent data suggest that an altered availability of cGMP could better elucidate the contrasting findings of intrahepatic vasoconstriction and peripheral systemic vasodilation than mere focus on NO availability. Preclinical and clinical data have demonstrated that targeting the NO-cGMP pathway in liver cirrhosis using PDE-5 inhibitors or sGC stimulators/activators decreases intrahepatic resistance through dilation of sinusoids, lowering portal pressure, and increasing portal venous blood flow. These results suggest further clinical applications in liver cirrhosis. Targeting the NO-cGMP system plays a role in possible reversal of liver fibrosis or cirrhosis. PDE-5 inhibitors may have therapeutic potential for hepatic encephalopathy. Serum/plasma levels of cGMP can be used as a non-invasive marker of clinically significant portal hypertension. This manuscript reviews new data about the role of the NO-cGMP signal transduction system in pathophysiology of cirrhotic portal hypertension and provides perspective for further studies.

## 1. Introduction

Portal hypertension (PH) is defined as portal venous pressure exceeding 5 mm Hg. Significant sequelae of PH occur when pressure exceeds 10–12 mmHg [1,2,3,4,5,6]. It develops gradually from asymptomatic abnormalities to life-threatening complications. The impairment of portal blood flow may occur in different anatomical locations. PH is classified as prehepatic (e.g., portal venous thrombosis), intrahepatic, and posthepatic (e.g., hepatic vein thrombosis, chronic right-sided heart failure) [2,7,8,9]. Intrahepatic PH is further differentiated into presinusoidal, sinusoidal, and postsinusoidal. Sinusoidal PH occurs most frequently as a direct consequence of liver cirrhosis and is the focus in this review. Successful therapy of PH aims to decrease the hepatic-venous pressure gradient (HVPG) by ≥20% of baseline or ≤12 mmHg [1,2,10,11]. A short-term response of HVPG to acute drug application (i.e., propranolol) by ≥10% of baseline or a reduction of ≤12 mmHg predicts a beneficial effect for prophylaxis of variceal bleeding [11,12,13,14].

An increase in hepatic vascular resistance to portal blood flow is the main factor of intrahepatic PH. The structural component contributing about 70% to the intrahepatic vascular resistance is caused by the disturbed liver architecture, fibrotic tissue, regenerative nodules, angiogenesis, narrowing of intrahepatic blood vessels by extracellular matrix proteins, and vascular occlusion. A dynamic component accounts for about 30%. It is caused by an impaired interplay of the sinusoidal cells, i.e., hepatocytes, liver sinusoidal endothelial cells (LSECs), and hepatic stellate cells (HSCs), characterized by transformation of contractile HSCs to myofibroblasts, synthesis of extracellular matrix proteins, and capillarization of LSECs. Eventually, this leads to an increased sinusoidal tone caused by an imbalance of dilating and constricting factors [7,15,16,17,18,19,20,21]. Moreover, elevated splanchnic blood flow due to dilation of splanchnic blood vessels contributes to the development and/or worsening of PH [21,22,23].

The main regulator of vascular and sinusoidal tone is the NO-cGMP pathway [19,20,24]. Endothelial NO-synthase (eNOS) generates NO from L-arginine, which diffuses to smooth muscle cells in peripheral blood vessels and HSCs in liver sinusoids. NO activates soluble guanylate cyclase (sGC), catalyzing the conversion of guanosine triphosphate (GTP) to cyclic guanosine monophosphate (cGMP). cGMP, in turn, activates protein kinases (e.g., protein kinase G, (PKG)), PDE-5, and cGMP-gated ion channels. cGMP induces a decrease of intracellular calcium ions and vascular dilation. cGMP is then converted to functionally inactive 5′-GMP by PDE-5 [25,26,27,28]. Among other regulatory pathways, the NO-cGMP signal transduction system is characteristically altered in liver cirrhosis. Hence, structural alterations and functional dysregulations play a crucial role in the pathogenesis of PH. For years, research in PH has been focused on NO and led to the theory of the “NO-paradox”—intrahepatic circulation is marked by an NO deficiency, in contrast to excess NO in the peripheral systemic circulation [29,30,31,32]. It states that vasoconstriction prevails in intrahepatic vasculature, whereas extrahepatic vasculature is characterized by dilation, leading to a hyperdynamic circulatory state.

This manuscript reviews recent studies on localization and expression of the enzymes eNOS, sGC, and PDE-5, as well as the molecule cGMP in a healthy and cirrhotic liver. It proposes shifting the focus to cGMP as the key component responsible for sinusoidal constriction and peripheral vasodilation in liver cirrhosis. We discuss study recommendations on the effects of sGC modulators and PDE-5-inhibitors in liver diseases such as modulation of the NO-cGMP pathway for treatment of portal hypertension [33], reversal of fibrosis/cirrhosis, therapy of hepatic encephalopathy using PDE-5-inhibitors, and determination of serum/plasma cGMP levels as an indicator of portal hypertension.

## 2. Current Research Directions in Components of the NO-cGMP Pathway in Experimental Liver Damage

NO and eNOS: In a healthy liver, eNOS is constitutively expressed. A number of studies showed reduced activity of eNOS in a cirrhotic liver. However, it is not clear whether this reflects reduced expression or disturbed post-translational modification of the enzyme [34,35,36,37,38,39,40]. Several more recent studies [41,42,43] found increased eNOS expression in a cirrhotic liver, however, this may depend on the stage or type of liver damage. Schaffner et al. (2018) [42] demonstrated upregulation of eNOS in TAA-induced rat liver fibrosis and cirrhosis and high expression of PDE-5. The same findings were reported by Uschner et al. (2020) [43]—eNOS was upregulated in both BDL-induced and CCl_4_-induced liver cirrhosis, accompanied by PDE-5 upregulation. In hepatic arteries of human cirrhotics, however, eNOS was upregulated and PDE-5 was downregulated, while in the aortic wall of cirrhotic rats, eNOS was upregulated, and PDE-5 was downregulated.

sGC: Using immunohistochemistry, Theilig et al. (2001) [44] detected sGC in HSCs and found decreasing amounts of this enzyme from the periphery of the hepatic lobule towards the central vein as an indicator of metabolic zonation of the NO-cGMP pathway. According to Davies et al. (2006) [45], sGC activity was lowered in BDL-induced rat liver cirrhosis. In contrast, Loureiro-Silva et al. (2006) [46], Lee et al. (2010) [47], Schwabl et al. (2018, 2020) [41,48], and Schaffner et al. (2018) [42] described overexpression of sGC in different types and stages of experimental liver damage. Hall et al. (2019) [49] described sGC in HSCs and myofibroblasts in a rat model of non-alcoholic steatohepatitis (NASH), however, it was not present in hepatocytes. In CCl4-induced rat liver cirrhosis, sGC was overexpressed and was also found in large amounts in fibrotic tissue within a cirrhotic liver. These findings were confirmed in human NASH. The sGC stimulator praliciguat markedly increased hepatic cGMP levels in experimental liver cirrhosis and reduced fibrosis and inflammation in a damaged liver [49]. Schwabl et al. (2018) described upregulation of the β1-subunit of sGC in BDL-induced liver cirrhosis and to a lesser extent in CCl4-induced liver cirrhosis, whereas the α1-subunit remained unchanged [41]. In a healthy rat liver, sGC was primarily detected in HSCs and hepatocytes, as well as in LSECs. After bile duct ligation, sGC expression was observed in HSCs, hepatocytes, and Kupffer cells. Stimulation of sGC by Riociguat decreased portal pressure and improved fibrosis, as demonstrated by histology and measurement of αSMA expression. Inflammatory markers were reduced in both types of experimental liver damage [48].

PDE-5: Perri et al. (2006) [50] described a defective cGMP-PKG pathway in activated HSCs contributing to an impaired NO-dependent response. Activity or expression of PDE-5 was not considered in this study. Loureiro-Silva et al. (2006) [46] found a decreased vasodilatory response to NO in experimental liver cirrhosis as a consequence of PDE-5 overexpression. Markedly increased protein expression of PDE-5 and a slight overexpression of sGCα1β1 were reported by Lee et al. (2010) [47] in BDL-induced liver cirrhosis. They observed that after a one-week administration of Sildenafil, sGC was further upregulated and PDE-5 was reduced. This effect was accompanied by a reduction of portal venous pressure. Schaffner et al. (2018) [42], on the other hand, found that PDE-5 is overexpressed in different stages of TAA-induced liver fibrosis and cirrhosis. This may override the overexpression of eNOS demonstrated by the same group. The description of a zonation of PDE-5 yielded a new aspect in the regulation of cGMP levels inside the liver—high sGC in the periphery of the liver lobule [44] stimulated cGMP production, whereas high PDE-5 in perivenular regions converted cGMP to inactive 5′-GMP before entering systemic circulation. In TAA-induced liver damage, this zonation was lost and apart from increased expression of PDE-5 in perisinusoidal cells, high amounts of this enzyme were detected in fibrotic liver tissue [42]. Uschner et al. (2020) [43] confirmed upregulation of eNOS and PDE-5 in BDL- or CCl4-induced rat liver cirrhosis. Consequently, intrahepatic cGMP was reduced. In contrast, in aortic wall tissue, PDE-5 was reduced and eNOS was upregulated in cirrhosis models. Corresponding results were obtained in cirrhotic livers of human transplant recipients: PDE-5 and eNOS were upregulated. However, in hepatic arteries, PDE-5 mRNA was reduced and eNOS mRNA was increased in comparison to those of healthy liver donors. Moreover, it was reported that eNOS was markedly expressed in LSECs, whereas PDE-5 was mainly found in HSCs. Udenafil treatment increased or normalized cGMP levels in a cirrhotic rat liver, inducing dilation of sinusoids and reduction of portal pressure, however, cGMP was further increased in aortic tissue of cirrhotic rats. The combination of udenafil and propranolol blunted the effect of udenafil alone on cGMP in aortic tissue, however, the dilating effect inside the liver was additive. Furthermore, eNOS and PDE-5 were reduced in aortic tissue after combined application of udenafil and propranolol. These data yielded the rationale for a combination therapy of cirrhotic portal hypertension using a PDE-5-inhibitor (dilation of sinusoids) and propranolol (constriction of splanchnic blood vessels by blockade of beta-2 adrenoreceptors). In addition, these data showed that peripheral vasodilation can be explained by increased cGMP due to peripheral PDE-5 downregulation. eNOS was not the main player in this circulatory disturbance.

Another study by Brusilovskaya et al. (2020) investigated the effects of tadalafil (a long-acting inhibitor of PDE-5), riociguat (a sGC-stimulator), and cinaciguat (a sGC-activator) in BDL-induced liver cirrhosis [48]. Tadalafil and riociguat decreased portal pressure by about 20% and 10%, respectively, and decreased intrahepatic vascular resistance with a minor effect on systemic circulatory parameters. Both tadalafil and riociguat reduced the extent of liver fibrosis in comparison to that in control and decreased serum levels of aminotransferases, suggesting a beneficial effect on liver damage. Cinaciguat did not show the same beneficial effects. After tadalafil and riociguat administration, hepatic smooth muscle actin alpha protein concentration decreased and expression of phosphorylated moensin was reduced. BDL did not lead to a decrease of intrahepatic cGMP content. However, both tadalafil and riociguat induced a marked increase of intrahepatic cGMP by 240% and 32%, respectively. This was the first study which compared the effects of targeting sGC or PDE-5. The results do not clearly favor any of the two.

cGMP: Niederberger et al. (1995) [51] confirmed high cGMP in aortic walls in experimental cirrhosis and hypothesized an increase in NO synthesis caused by eNOS. The authors demonstrated that a correction of increased NO production normalized hyperdynamic circulatory disturbance [52]. Tahseldar et al. (2009) [53] investigated activities of PDEs in kidney and liver of rats with cirrhosis. They found an increase of cGMP hydrolyzing activity by 24% in cirrhosis (mainly PDE-5). It was observed that while PDE-5 was increased in mesenteric arteries, it remained unchanged in the aorta of cirrhotic animals. The study conducted by Uschner et al. (2020) [43] clarified some contrasting data about the disturbance of the NO-cGMP pathway in peripheral blood vessels in cirrhosis—PDE-5 was found to be downregulated in the aortic wall in experimental cirrhosis and in hepatic arteries in human cirrhosis. These data are consistent with the increase of serum cGMP irrespective of eNOS activity.

## 3. Role of the NO-cGMP Pathway in PH Pathophysiology

Considering these recent findings, we propose explanations for these controversial findings and thereby offer an alternative perspective in understanding the pathophysiology of PH. We suggest a paradigm highlighting how the NO-cGMP pathway is altered in liver cirrhosis. Modulation of sGC and/or PDE-5 regulates sinusoidal cGMP availability, thus influencing sinusoidal tone, portal pressure, and portal venous blood flow (Figure 1).

NO is generated by eNOS in sinusoidal endothelial cells and diffuses into the neighboring hepatic stellate cells, where it binds to and activates sGC. This enzyme catalyzes the conversion of GTP to cGMP. This intracellular second messenger, in turn, triggers distinct downstream signaling effects (e.g., lowering of intracellular Ca ions) which eventually exert vasodilation. As a negative feedback mechanism, rising cGMP concentrations initiate the activation of PDE-5, which mediates cGMP inactivation. The opposing zonation of sGC and PDE-5 may suggest regulation of sinusoidal cGMP. It is synthesized in the periphery of the sinusoids (high sGC concentration), exerts its physiological function along the sinusoids and is inactivated before it enters systemic circulation (high PDE-5 concentration in zone 3 hepatocytes). Figure 1, panel A.

In liver cirrhosis, both eNOS and sGC are overexpressed. Their activity is overridden by the marked overexpression of PDE-5 primarily in perisinusoidal cells and the loss of the physiological zonation of PDE-5. The final dilator cGMP is converted to inactive 5′-GMP. Sinusoids persist in a constricted state. Figure 1, panel B.

PDE-5 is markedly overexpressed in liver cirrhosis, however, its activity can be blocked by PDE-5 inhibitors. This leads to a normalization of cGMP concentrations and, consequently, sinusoidal dilation. Figure 2, left side.

Comparable effects are achieved if sGC activity is further enhanced by either an sGC stimulator or activator. This overrides the overexpression of PDE-5, leading to normalization of cGMP levels and sinusoidal dilation. Figure 2, right side.

Further studies should show whether the etiology of liver damage has an effect on these changes in the NO-cGMP pathway.

Figure 3 shows the distribution of PDE-5 in perisinusoidal cells and the expression in zone 3 hepatocytes in a healthy human liver. Immunolabeling of cirrhotic liver tissue highlights the presence of PDE-5 in fibrous septa, in hepatocytes adjacent to veins, and in perisinusoidal cells scattered throughout the parenchymal nodules.

Targeting the components of the NO-cGMP pathway in cirrhotic liver management using PDE-5 inhibitors or sGC modulators could lower portal pressure and increase portal perfusion. These effects might have clinical consequences.

These drugs are potential therapeutic alternatives for portal hypertension.PDE-5 inhibitors or sGC modulators may be useful for reversal of liver fibrosis/cirrhosis, as shown in animal studies.PDE-5 inhibitors are a promising therapy of hepatic encephalopathy in liver cirrhosis.Serum levels of cGMP can be used as a simple non-invasive marker of clinically significant portal hypertension.

## 4. Modulation of the NO-cGMP Pathway in a Healthy and Cirrhotic Liver

### 4.1. Effect in Portal Hypertension

The effects of PDE-5 inhibitors on portal hemodynamics in experimental models of liver damage were investigated by several authors [41,42,43,47,48,54,55]. Consistently, a decrease of portal pressure was demonstrated, as well as an increase of sinusoidal flow or portal blood flow. PDE-5 inhibitors were also tested in the clinical setting by several teams of researchers [56,57,58,59,60,61,62]. A decrease in HVPG of >10% in acute testing, indicating a beneficial clinical effect, was proven in the majority of studies [54,56,57,59,61]. For details of the studies, see Table 1.

Preclinical and clinical data provide evidence that targeting sCG (stimulators or activators) or PDE-5 inhibitors in liver cirrhosis lowers portal pressure and intrahepatic resistance, thereby increasing sinusoidal blood flow. Thus, PDE-5-inhibitors and sGC stimulators/activators can be used as a novel therapy in treating portal hypertension. The data also suggest that although the effects of sGC and PDE-5 inhibitors are dependent on NO synthesis, eNOS activity it not the limiting factor. The main regulator of sinusoidal tone, cGMP, can be modulated by targeting sGC and/or PDE-5.

The results of preclinical and clinical studies are summarized in Table 1. For further details, we refer to Kreisel and Schaffner et al. (2020) [33].

### 4.2. Targeting the NO-cGMP Pathway May Contribute to Reversal of Liver Fibrosis/Cirrhosis

Recent clinical and experimental studies suggested that chronic liver disease and resulting portal hypertension may be at least partially reversible [64,65,66,67]. Zoubek et al. (2017) [68] summarized the aims of antifibrotic therapy as follows: “The fundaments of fibrosis resolution rely on four cardinal points: discontinuation of the primary cause of chronic hepatic injury; regression of myofibroblasts from an activated to a deactivated status and/or their elimination; degradation of excessive matrix; switch of proinflammatory milieu to a restorative environment”.

For example, regression of fibrosis/cirrhosis has been observed in alcoholic liver disease after cessation of alcohol consumption [69,70], in chronic HBV infection [71,72,73] after long-term antiviral therapy, in chronic HCV infection after elimination of the virus with direct antiviral agents [65,66,74,75,76], and in non-alcoholic fatty liver disease after bariatric surgery [77,78]. Influencing the gut microbiome may play an important role in the treatment of NAFLD [79].

The cellular key players in the development of liver fibrosis are HSCs, which differentiate into myofibroblasts. The discovery of novel pathways and mediators revealed the plasticity and complexity of HSC activation but also yielded information about potential approaches to reverse HSC activation [80]. Reversal of fibrosis is accompanied by apoptosis of myofibroblasts or redifferentiation into inactivated HSCs. Elimination of HSCs is achieved via apoptosis or senescence [81,82]. According to Kisseleva et al. (2021) [83], several drugs are under consideration for therapy of liver fibrosis that interfere with HSC activation, such as PPARy agonists, LPAR1 agonists, NOX inhibitor, and hedgehog inhibitors, among others. Thus, it is worth investigating whether drugs which modulate the NO-cGMP pathway, influencing the function of HSCs, may be capable of reversing liver damage. Selicean et al. (2021) [67] reviewed pharmacological approaches to address portal pressure in the regression of chronic liver disease, such as ACE-inhibitors or ARBs, statins, the FXR agonist obeticholic acid, the Rho-kinase inhibitor fasudil, endothelin receptor antagonists, and the multikinase inhibitor sorafenib. The authors mentioned therapy with inhibitors of PDE-5 as a further possibility.

Furthermore, Knorr et al. (2011) [84] demonstrated that the sGC activator BAY 60-2770 ameliorated fibrosis in the rat model of BDL-induced cirrhosis independent of NO. Choi et al. (2009) [55] showed that the PDE-5 inhibitor udenafil decreased portal pressure by about 30%, reduced the degree of fibrosis, and suppressed the expression of procollagen type I and alpha-smooth muscle actin mRNA as an indicator for reduced HSC activity. According to Xie et al. (2012) [85], the sGC activator BAY 60-2770 accelerated the complete reversal of capillarization (restored differentiation of LSECs) in TAA-induced liver damage. Restoration of differentiation to LSECs led to quiescence of HSCs and regression of fibrosis even in the absence of further exposure to the sGC activator.

In a NASH model investigated by Flores-Costa et al. (2018) [86], sCG and cGMP were both lowered. These were restored to normal levels by the sGC stimulator IW-1973. This had a protective effect on hepatic steatosis, inflammation, and fibrosis. Similarly, as demonstrated by Hall et al. (2019) [49] in CCl4-induced liver cirrhosis, the sGC stimulator praliciguat markedly increased hepatic cGMP levels and reduced fibrosis and inflammation in the damaged liver. Schwabl et al. (2018) [41] reported that stimulation of sGC by riociguat decreased portal pressure and also improved fibrosis as demonstrated by histology and measurement of α-smooth muscle actin expression in BDL-induced and CCl4-induced liver cirrhosis. Inflammatory markers were reduced in both types of experimental liver damage. In the study of Brusilovskaya et al. (2020) [48], the long-acting PDE-5 inhibitor tadalafil and the sGC stimulator riociguat decreased portal pressure and intrahepatic vascular resistance and diminished the extent of fibrosis in BDL-induced cirrhosis.

Preclinical data suggest that modulation of the NO-cGMP pathway may induce or accelerate reversal of fibrosis/cirrhosis.

### 4.3. Role of Inhibitors of PDE-5 or Modulators of sGC in Hepatic Encephalopathy

The pathogenesis of hepatic encephalopathy (HE) is a complex process involving a disturbed interplay of the gut, liver, kidneys, skeletal muscles, and brain [87,88,89,90,91,92,93]. The common denominator is hyperammonemia. Therapy of HE [94,95,96,97,98] aims to correct underlying and precipitating factors—treatment of viral infections, cessation of offending drugs or alcohol, management of bleeding episodes, prevention of malnutrition or electrolyte imbalance, correction of intestinal dysbiosis, therapy of portosystemic shunts which bypass portal blood from the liver, and improvement of liver perfusion. The most promising therapeutic approaches target hyperammonemia [88,91,93,97,99,100].

Ammonia is produced mainly from protein digestion in the gastrointestinal tract and, in part, from glutamine via glutaminase in the kidney [101]. In healthy individuals, the ammonia level is stable and can be modified by urea synthesis in periportal hepatocytes and/or synthesis of glutamine in perivenous hepatocytes in the liver. Although hyperammonemia has been considered the main factor responsible for neurological changes, other factors like inflammation, inflammatory cytokines, disturbed cerebral blood flow, hyperthermia, hyponatremia, oxidative stress, catabolic state in liver cirrhosis, and increased levels of bile acids and lactate may also be involved. Blood ammonia levels do not always correlate with the severity of encephalopathy. On the other hand, biochemical changes observed in HE can be reproduced by increasing blood or brain ammonia levels in experimental animals. In cultured astrocytes, exposure to ammonium salts reproduces the morphological and biochemical findings observed in HE [102,103,104,105]. Hyperammonemia leads to an increase of glutamine in astrocytes that causes hypertonicity, cytotoxic astrocyte swelling, and brain edema [105,106,107]. These astrocytic functional changes impair the glutamate-glutamine metabolism, leading to excitotoxicity and neuronal dysfunction. Damage to astrocyte-neuronal communication as well as the impairment of glutamatergic and GABAergic neurotransmission further compromise cerebral function and may explain neurological deficits of HE.

The glutamate-NO-cGMP pathway, including N-methyl-D-aspartate (NMDA) receptors, plays important roles in function and intercellular communication. Several investigators demonstrated modifiable learning ability depending on extracellular cGMP levels in rats [102,103,104].

Hyperammonemic rats show impaired function of the glutamate-NO-cGMP pathway, reduced levels of extracellular cGMP in the cerebellum, and reduced ability for spatial learning and memory [108]. Treatments that increase cGMP in the cerebellum were observed to restore learning in rats with hyperammonemia or hepatic encephalopathy [102,103,104,105].

These studies suggest that in pathological situations wherein reduced extracellular cGMP concentration is associated with reduced function of the glutamate-NO-cGMP pathway, it is possible to improve the mechanisms of such a pathway and enhance learning ability using treatments that increase cGMP levels. This may be achieved by administering phosphodiesterase inhibitors such as zaprinast or sildenafil [102,106], anti-inflammatory drugs like ibuprofen, inhibitors of MAP-kinase p38 [107,109], or compounds reducing GABAA receptor activation [108,110,111]. Continuous intracerebral administration of zaprinast, an inhibitor of phosphodiesterase, increased extracellular cGMP and restored learning ability in hyperammonemic rats. In rats with porto-caval anastomosis or hyperammonemia, sildenafil normalized the function of the glutamate-NO-cGMP pathway and extracellular cGMP in the brain in vivo [112]. Other research groups showed that administration of sildenafil or vardenafil improved the memory performance of hyperammonemic rats in an object recognition task [111], and sildenafil improved long-term retention of an inhibitory avoidance response in mice [113].

The memory-enhancing effects of PDE-5 inhibition in experimental studies may possibly be related to increased blood flow and, consequently, increased glucose metabolism, as PDE-5 inhibitors are known vasodilators [112]. Biochemical and immunohistochemical studies support the effect of PDE-5 inhibition on the glutamate-NO-cGMP pathway as the mechanism of action. Additionally, sildenafil was shown to increase axon and myelin density and modulate gene expression in rats with multiple microinfarction, as well as decrease inflammatory responses in aged rats [114]. This suggests that the mechanisms involved go beyond a mainly vasomotor mechanism of action. Figure 4 describes the disturbance of the NO—cGMP pathways in hepatic encephalopathy.

The influence of PDE-5 inhibitors on cerebral function in HE is complex. In addition to modifying the glutamate-NO-cGMP pathway, changes in cerebral blood flow may also play a role. Influencing inflammation via intracellular cGMP concentration is also conceivable. The increase in sinusoidal perfusion, decrease of portal pressure, and reduction of shunt volume might lead to an improvement in the detoxification capacity of the liver. Further clinical studies are needed to prove the effect of PDE-5 inhibitors on hepatic encephalopathy.

### 4.4. Plasma/Serum cGMP as a Potential Marker of Clinically Significant Portal Hypertension (CSPH)

Numerous blood tests have been established to recognize or differentiate various liver diseases [115,116,117,118]. Several biochemical parameters in liver cirrhosis indicate hepatocyte damage or cholestasis, while others indicate stimulated synthesis of extracellular matrix proteins (laminin, hyaluronic acid, procollagen-III-peptide, Mac-2-binding protein glycosylation isomer) by stimulated HSCs (myofibroblasts), thus reflecting the degree of liver fibrosis. Various serum markers are suitable tools to detect clinically significant portal hypertension [119,120,121,122,123].

Serum/plasma cGMP could be a potential indicator of clinically significant portal hypertension in which the pathophysiological background is based on the opposed disturbance of the NO-cGMP pathway inside (low cGMP) compared to outside the cirrhotic liver (high cGMP). Inside the cirrhotic liver, cGMP is converted into biologically inactive 5′-cGMP by overexpressed PDE-5. This results in low cGMP in sinusoids and in liver outflow [42]. In contrast, in peripheral arteries, PDE-5 is underexpressed in liver cirrhosis [43]. This leads to high cGMP levels in the peripheral circulatory system. The disturbance of the NO—cGMP pathway in the peripheral circulation is shown in Figure 5.

In patients with progressed liver cirrhosis, similar findings were reported. Kirstetter et al. (1997) [124] described elevated plasma cGMP levels in patients with progressed cirrhosis. Levels of ANP (a stimulator of membrane bound guanylate cyclase) were unchanged, while sGC was not measured. Montoliu et al. (2005) [125] described increased levels of plasma cGMP, as well as nitrates and ANP, in human liver cirrhosis. To further explain these data, increased stimulation of sGC by NO and membrane-bound GC by ANP as well as enhanced release of intracellular cGMP were discussed. The same authors (2010) demonstrated increased plasma cGMP levels in patients with HE [126] in a separate study. Similarly, Siqueira et al. (2008) [127] described significantly increased plasma levels of cGMP in patients with alcoholic liver cirrhosis. cGMP levels were negatively correlated with the MELD score, suggesting that progression of cirrhosis leads to higher plasma cGMP levels. According to Felipo et al. (2014) [128], patients with cirrhosis and minimal HE displayed increased blood ammonia, cGMP, IL-6, IL-18, and 3-nitrotyrosine. Sturm et al. (2021) [129] also published preliminary data that plasma cGMP was found to be significantly elevated in patients with clinically significant portal hypertension (CSPH), whereas in cirrhotics without CSPH, cGMP remained normal. These data suggest that plasma/serum cGMP might be an indicator of CSPH.

## 5. Conclusions

The “NO-paradox” states that NO availability inside the liver is reduced and NO production in the peripheral systemic circulation is increased, leading to vascular disturbance in liver cirrhosis. However, recent data suggest that altered availability of cGMP better elucidates the contrasting findings of intrahepatic vasoconstriction and peripheral systemic vasodilation. We suggest a focus shift from NO to cGMP and propose the alternative “cGMP-paradox”. The enzymes sGC and PDE-5 play a crucial role in the regulation of local cGMP levels. Hence, targeting these enzymes by PDE-5 inhibitors or sGC stimulators/activators could induce dilation of hepatic sinusoids, decrease intrahepatic resistance, increase portal venous blood flow, and consequently, lower portal pressure in cirrhotic portal hypertension. These effects suggest further clinical applications in liver cirrhosis. Inhibitors of PDE-5 or stimulators/activators of sGC may contribute to reversal of liver fibrosis or cirrhosis. Inhibitors of PDE-5 may have therapeutic potential for hepatic encephalopathy. Finally, plasma/serum levels of cGMP can be used as non-invasive marker of clinically significant portal hypertension.

## Figures and Tables

**Figure 1 ijms-22-10372-f001:**
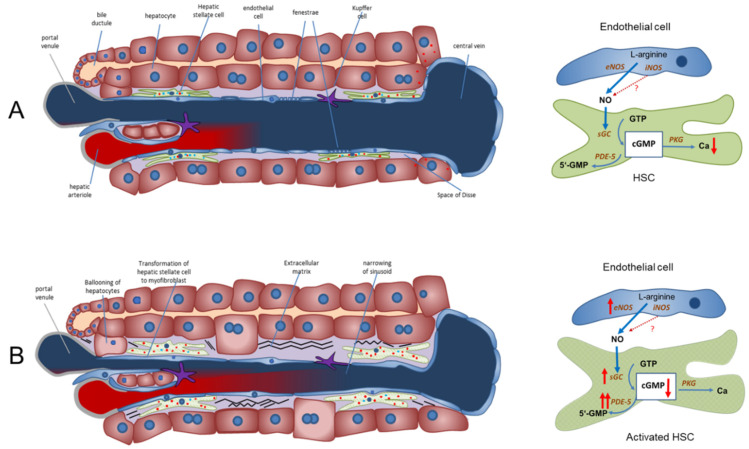
The NO-cGMP pathway as the main regulator of sinusoidal and vascular tone in a healthy and cirrhotic liver (modified from Schaffner et al. (2018) [42] and Kreisel and Schaffner et al. (2020) [33]). (**A**) Regulation of sinusoidal tone in a healthy liver. In a healthy liver, an opposing zonation of sGC and PDE-5 may lead to high cGMP production in the peripheral parts of the hepatic lobule (high sGC), in which cGMP may exert its physiological function inside the sinusoids. Excess cGMP is possibly degraded by a high presence of PDE-5 (zone 3 hepatocytes) before entering extrahepatic vasculature. (**B**) Disturbed regulation of sinusoidal tone in liver cirrhosis. Altered expression of key enzymes in the NO-cGMP pathway (overexpression of eNOS and sGC and marked overexpression of PDE-5 in perisinusoidal cell, mainly activated HSCs) leads to reduced cGMP concentrations and sinusoidal constriction, increasing portal pressure. ↑ increased expression; ↑↑ markedly increased expression; cGMP ↓ decreased concentration. Red dots: PDE-5. Blue dots: sGC.

**Figure 2 ijms-22-10372-f002:**
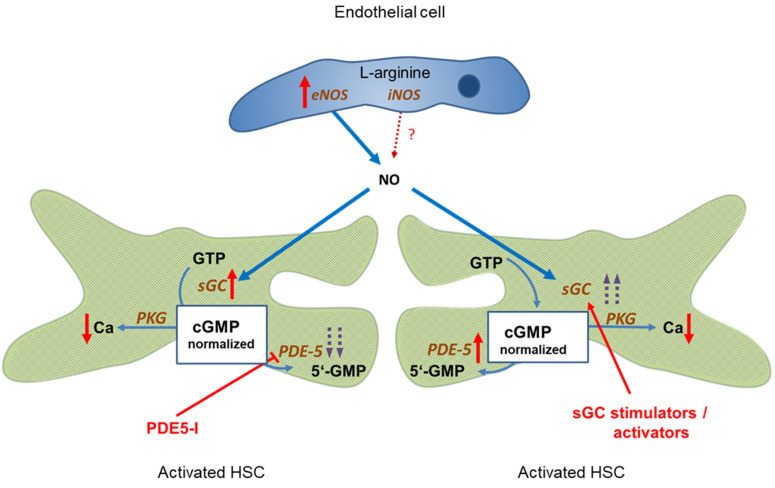
Targeting the NO-cGMP pathway for therapy of portal hypertension. Left side: Administration of PDE-5 inhibitors in liver cirrhosis leads to normalization of cGMP concentration and sinusoidal dilation. Portal pressure decreases and portal blood flow increases. Right side: Enhancing sGC activity in liver cirrhosis using an sGC stimulator or activator leads to increased cGMP levels and decreased sinusoidal tone. Portal pressure decreases and portal blood flow increases. ↑ increased expression; ↑↑ markedly increased expression; Ca ↓ decreased concentration; 
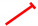
 inhibition; 
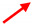
 stimulation/activation.

**Figure 3 ijms-22-10372-f003:**
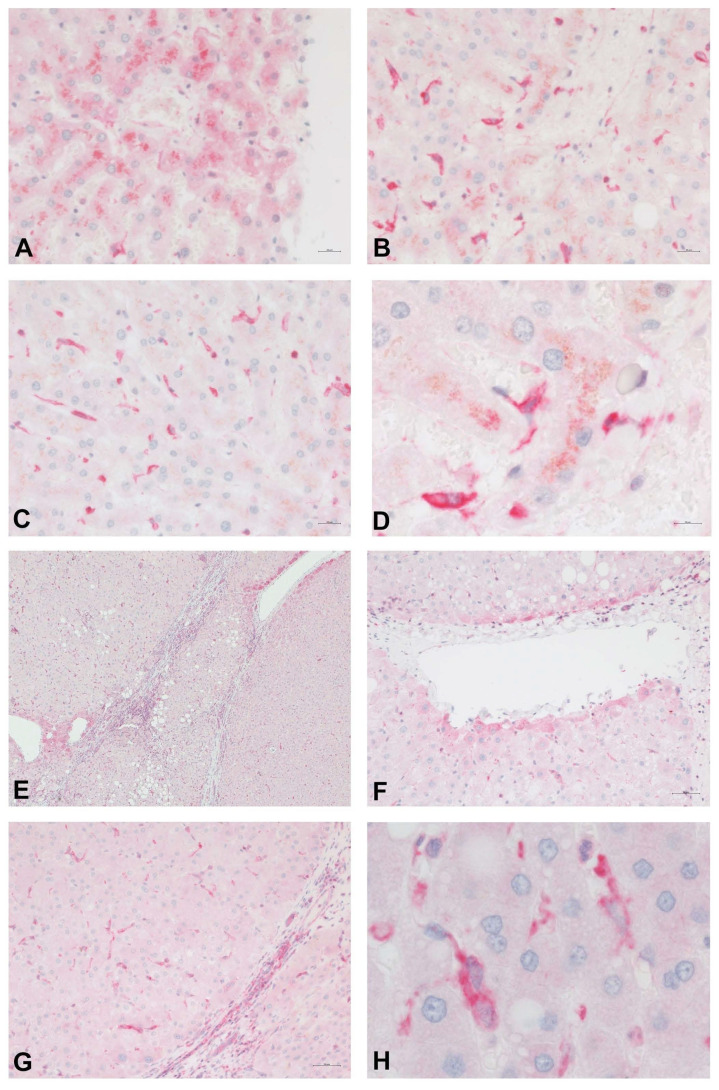
Immunostain for PDE-5 (red) of normal (**A**–**D**) and cirrhotic (**E**–**H**) human liver tissue. In normal control liver samples, PDE-5 is weakly expressed in hepatocytes around terminal hepatic venules (**A**). A higher intensity of staining is noted in perisinusoidal cells, especially in the perivenular region (**B**–**D**). Immunolabeling of cirrhotic liver tissue highlights the presence of PDE-5 in fibrous septa (**E**,**C**), in hepatocytes adjacent to veins (**E**,**F**), and in perisinusoidal cells scattered throughout the parenchyma.

**Figure 4 ijms-22-10372-f004:**
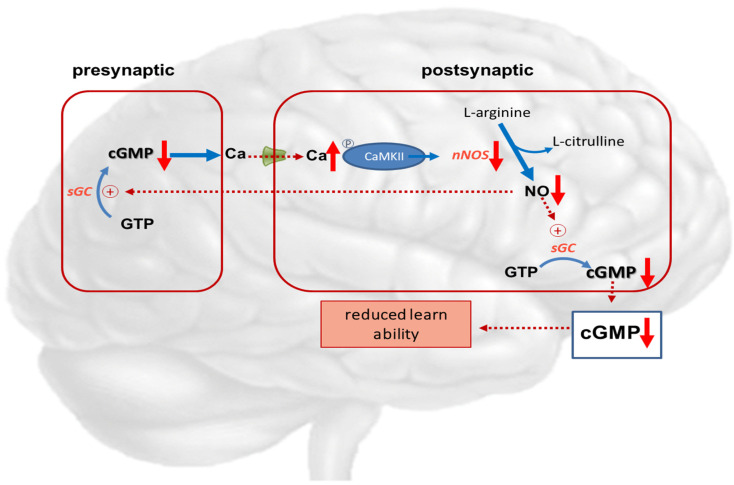
Potential role of cGMP in hepatic encephalopathy. Hyperammonemia reduces the function of the glutamate-nitric oxide-cGMP pathway. Activation of NMDA receptors leads to increased intracellular calcium in the postsynaptic neuron. Calcium binds to calmodulin, which in turn activates function of calcium/calmodulin-dependent protein kinase II (CaMKII) by phosphorylation. CaMKII itself phosphorylates neuronal NO-synthase, reducing its activity and NO formation. This results in reduced synthesis of cGMP, leading to reduced learning ability. 
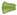
 NMDA-receptor; ↑ increased expression; ↑↑ markedly increased expression; ↑ increased concentration; ↓ decreased concentration.

**Figure 5 ijms-22-10372-f005:**
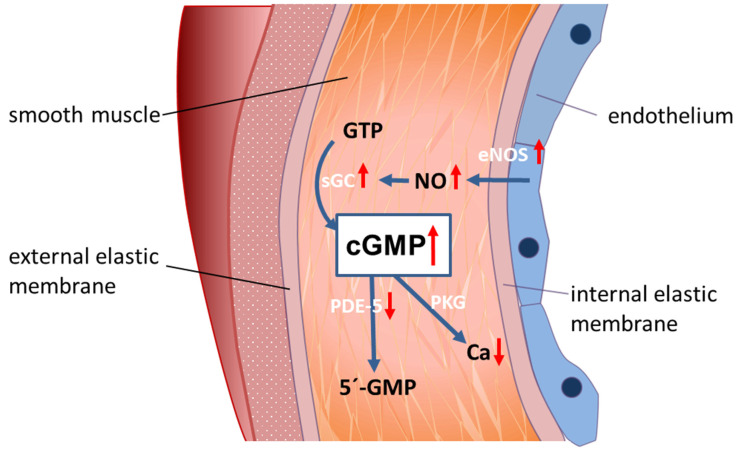
A model for peripheral vasodilation in progressed liver cirrhosis. In liver cirrhosis, NO is overexpressed in peripheral arterial blood vessels, while PDE-5 is decreased. This leads to high cGMP concentration and peripheral vasodilation.

**Table 1 ijms-22-10372-t001:** Effect of PDE-5 inhibitors and sGC modulators on portal pressure, Kreisel and Schaffner et al. (2020) [33].

	Model	Compound	Dosage and Route	ΔMAP	ΔPVP	Remarks
Colle 2004 [63]	Wistar rats, BDL	Sildenafil	0.01–10 mg/kg, i.v.	−1–20% even more in sham rats	+2– +6% even more in sham rats	
Halverscheid 2009 [54]	Sprague-Dawley rats, non-cirrhotic	Sildenafil Vardenafil	1, 10, or 100 µg/kg, i.v. 1, 10, or 100 µg/kg, i.v.	1.1; −3.9; −2.6% −11.0; −8.7; −7.4%	In all groups no increase, but decrease over time	3.3; 24.1; 18.3% 15.9; 29.2; 23.9% increase in portal flow
Schaffner 2018 [42]	Wistar rats, TAA	Sildenafil	0.1–1.0 mg/kg, i.v.	−14–−17%	−13–−19%	
Uschner 2020 [43]	Sprague-Dawley rats, BDL or CCL_4_	Udenafil Udenafil/propranolol Udenafil 1 or 5 mg/kg	1 or 5 mg/kg 1 mg/kg	1 mg/kg: −20%; 5 mg/kg: −22% −7.5% 1 mg/kg: −31%; 5 mg/kg: −34%	−30–−23% −40% −30–−0%	Significant reduction of portal pressure
Lee 2010 [47]	Sprague-Dawley rats, BDL	Sildenafil, 1 week	0.25 mg/kg twice daily p.o.		−25%	
Choi 2009 [55]	Sprague-Dawley rats, BDL	Udenafil for 3 weeks	1, 5, or 25 mg/kg p.o.		−14, −13, −31%	
Deibert 2006 [56]	Human, cirrhotic (*n* = 18)	Vardenafil	10 mg, p.o.		−19% (*n* = 5)	Hepatic arterial resistance and portal flow increased significantly
Bremer 2007 [57]	Human, cirrhotic PPHTN (*n* = 1)	Tadalafil	10 mg, p.o.		−30%	Pulmonary arterial pressure −25%
Lee 2008 [58]	Human, cirrhotic (*n* = 7)	Sildenafil	50 mg, p.o.	Unchanged	+1%	Pulmonary arterial and sinusoidal resistance significantly reduced
Clemmesen 2008 [59]	Human, cirrhotic (*n* = 10)	Sildenafil	50 mg, p.o.	−14%	−11%	
Tandon 2010 [60]	Human, Cirrhotic (*n* = 12)	Sildenafil	25 mg, p.o.	−8%	−4% n.s.	Dose of Sildenafil too low
Kreisel 2015 [61]	Human, cirrhotic (*n* = 30)	Udenafil	12.5; 25; 50; 75; 100 mg p.o. acute 6 days		−3.5; −4.5; −7.5; −25.1; −17.3%−14.4; 3.1; −14.0; −13.5; −16.8%	Significant reduction of HVPG with ≥75 mg Udenafil in the acute setting and after 6 days
Deibert 2018 [62]	Human, cirrhotic (*n* = 1)	Vardenafil Tadalafil	10 mg 5 mg	−11%	−14% −15%	Sustaining reduction of HVPG > 10 months
Schwabl [41]	BDL rats, CCl_4_, BDL	Riociguat			−20% in early cirrhosis	Improvement offibrosis
Brusilovskaya [48]	BDL cirrhosis	Tadalafil Riociguat Cineciguat	1.5 mg/kg bw 0.5 mg/kg bw 1 mg/kg bw		−20% −40% no effect	cGMP + 40% cGMP + 239%

## Data Availability

The original data are available in the cited references.

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
