# Peer review of "Cyclic GMP in Liver Cirrhosis—Role in Pathophysiology of Portal Hypertension and Therapeutic Implications"

_ijms, 2021, doi:10.3390/ijms221910372_

Round 1

Reviewer 1 Report

The manuscript provides important new details in translation work on portal hypertension. In my opinion, its structure, comprehensiveness and value is obvious. 

Author Response

We thank the reviewer for his comment

Reviewer 2 Report

The Authors have fairly answered my previous comments. 

Author Response

We thank the reviewer for his comment.

Reviewer 3 Report

After revise, it has significant improvement. I have the last concern regarding the manuscript.

  1. When the abbreviation PDE-5-Is appears in the paper at the 1st time, please write out its whole name PDE-5-inhibitors.

Author Response

We thank for this comment. We did as the reviewer suggested.

This manuscript is a resubmission of an earlier submission. The following is a list of the peer review reports and author responses from that submission.

Round 1

Reviewer 1 Report

How to decrease portal hypertension and increase blood supply is the key to treat liver fibrosis and cirrhosis. In this review, authors introduced the NO-cGMP signal pathway, explained the function of key factors, such as eNOS, sGC, PDE-5 and cGMP, between LSECs and HSCs in detail, meanwhile compared them with the function in peripheral blood wall. On the other hand, authors introduced the function of PDE-5-inhibitors and sGC stimulators in animal liver fibrosis model and clinical treatment, the documents have been calculated and listed in table. Diagrams have been used to show the mechanism of NO-cGMP signal pathway. Authors found maybe PDE-5 inhibitors have therapeutic potential for hepatic encephalopathy. It is a quite interesting review; however, I have a few concerns regarding the manuscript.

  1. Please explain why Serum/plasma levels of cGMP can be used as non-invasive marker of clinically portal hypertension.
  2. The word in fig 1 and 2 is too small to read.
  3. Label “A” and “B” in Fig 2 left and right pictures.
  4. Page 3, line 104, change “porta” to “portal”

Author Response

Response to Reviewer 1

We thank the Reviewer for thoroughly reading the manuscript and for the insightful suggestions.

  1. In paragraph “4. Plasma/serum cGMP as potential marker of clinically significant portal hypertension (CSPH),” we tried to explain why serum/plasma cGMP can be used as a marker of clinically significant portal hypertension as follows:

Inside the cirrhotic liver, cGMP is converted into biologically inactive 5’-cGMP by overexpressed PDE-5. This leads to low cGMP in sinusoids and in hepatic venous outflow. As shown in our previously published paper (Schaffner et al. 2018), cGMP can be increased by the application of a PDE-5-inhibitor. The same result was demonstrated by other researchers. In contrast, in peripheral arteries PDE-5 is underexpressed in liver cirrhosis which leads to high cGMP levels in the peripheral circulatory system.  Additionally, several investigators have already demonstrated this effect: Increased peripheral cGMP levels in progressed human liver cirrhosis compared to healthy liver.

We included a short paragraph to better explain the rationale in the text.

  1. We agree with the reviewer on this point and improved the text accordingly.
  2. We included “A” and “B” for left and right panel. This was omitted in our first version.
  3. We corrected this typing error.

Reviewer 2 Report

This was a well-written review on nitrix-oxide (NO) pathway on the pathogenesis of portal hypertension. The paper focused on the NO paradox, namely the different expression of the NO pathway in the liver and in the splanchnic circulation in cirrhosis.

The Authors reviewed the latest studies reporting innovative data on this field. They then proposed a shift from NO to the cGMP expression as the main driver of this altered pathway. Finally, they highlighted the role of such a pathway also in hepatic encephalopathy.

My comments:

  • The Authors reported some data about the expression of sGC both in humans and animal models with NASH. Does the underlying etiology of liver disease play a significant role in such expression?
  • The Authors described the "NO paradox" throughout the manuscript, namely the reduced expression in the cirrhotic liver and its over-expression in the peripheral arterial blood vessels. Then, they proposed the use of serum levels of cGMP as a future biomarker of portal hypertension. How can we balance the liver underexpression and the peripheral overexpression when we measure serum cGMP?
  • The Authors reported some pilot studies where sildenafil or other PDE-5 inhibitors were investigated as potential drugs for reducing portal hypertension in patients with cirrhosis (refs. 56-62). In my opinion, a better description of these studies (including different dosages, side effects) would be of help for the Reader.

- Minor comments: there are some typos throughout the manuscript (e.g., porta instead of portal hypertension; conversation instead of conversion).

Author Response

Response to Reviewer 2.

We thank the Reviewer for the comments and helpful suggestions.

  1. At present, it is not yet clear whether or not the etiology of liver damage has an influence on the disturbed expression of key enzymes in the NO-cGMP signal system. We added a sentence in the paragraph “Role of NO-cGMP pathway in PH pathophysiology” and hope that our manuscript would inspire further studies on this topic.
  2. This manuscript reviews data which suggest that a disturbed expression of eNOS or disturbed bioavailability of NO less likely to be the main driver in cirrhotic portal hypertension, but rather the opposed cGMP bioavailability. The constricted state of sinusoids results from low cGMP (overexpression of PDE-5), while peripheral blood vessels are dilated due to increased cGMP, which in turn results from underexpression of PDE-5. Previous preclinical and clinical data show that there is a correlation between serum/plasma cGMP levels in the peripheral blood and clinically significant portal hypertension. Current data yielded a logical explanation for this observation. However, further studies have to be performed to elucidate more details of this correlation and to evaluate whether a ratio of cGMP concentrations in portal vein, hepatic vein, or peripheral veins is useful.
  3. The Reviewer asks for more details about the effects of PDE-5 inhibitors or sGC modulators on cirrhotic portal hypertension. At the end of the paragraph “Modulation of the NO – cGMP pathway in healthy and cirrhotic liver: 1. Effect in portal hypertension” we added the following sentences:

The results of preclinical and clinical studies are summarized in Table 1. For further details, we refer to Kreisel and Schaffner et al. (2020) [33].

  1. We corrected the typing errors.

Reviewer 3 Report

Dear authors and Editors, thank you for the opportunity to review that significant overview paper on the potential role of cGMP metabolism on portal hypertension caused by cirrhosis. The manuscript is comprehensive, interesting and fine illustrated. In my opinion, only minor correction on English language style are needed. 

Author Response

Response to Reviewer 3

We thank the Reviewer for his insightful and friendly remarks. We corrected all typing errors.

Round 2

Reviewer 2 Report

The Authors fairly answered my previous comments.